# Epidemiology of Musculoskeletal Injuries in Golf Athletes: A Championship in Portugal

**DOI:** 10.3390/ijerph21050542

**Published:** 2024-04-25

**Authors:** Beatriz Minghelli, Ana Sofia Palma Soares, Carolina Duarte Cabrita, Claudia Coelho Martins

**Affiliations:** 1School of Health Jean Piaget Algarve, Piaget Institute, 8300-025 Silves, Portugal; 57001@ipiaget.pt (A.S.P.S.); 57477@ipiaget.pt (C.D.C.); 57107@ipiaget.pt (C.C.M.); 2INSIGHT—Piaget Research Center for Ecological Human Development, Av. João Paulo II, Lote 544, 2º Andar, 1950-157 Lisbon, Portugal

**Keywords:** incidence, golf, injuries, prevalence

## Abstract

Although golf is a low-impact sport without physical contact, its movements are carried out over a large range of motion, and their repetition can predispose athletes to the development of injuries. This study aimed to investigate the epidemiology of musculoskeletal injuries in golf athletes who participated in championships in southern Portugal, determining the types, locations and mechanisms of injury and their associated risk factors. The sample consisted of 140 athletes aged between 18 and 72 years, 133 (95%) being male. The measuring instrument was a questionnaire about sociodemographics, modality and injuries’ characteristics. Throughout golf practice, 70 (50%) athletes reported injuries, totaling 133 injuries. In the 12-month period, 43 (30.7%) athletes suffered injuries, totaling 65 injuries. The injury proportion was of 0.31, and the injury rate was of 0.33 injuries per 1000 h of golf training. The most common injury type was muscle sprain or rupture (19; 30.9%), located in the lumbar spine (17; 27%), in which the repetitive movements were the main injury mechanism (42; 66.7%). The athletes who trained 4 times or more per week were 3.5 more likely (CI: 0.97–12.36; *p* = 0.056) to develop an injury while playing golf. Moderate injury presence was observed, with the high training frequency being an associated risk factor.

## 1. Introduction

Golf is played in 206 countries across the world by around 60 million people of all ages. Its global reach is also evidenced by its reinsertion in the 2016 Olympic Games [1]. According to the 2016 U.S. Golf Economy Report, golf’s direct impact on the economy was approximately US$ 84.1 billion, which represents an increase of US$ $68.6 billion compared to 2011 [2].

Golf requires coordinated movement of muscles throughout a high range of motion and, considering the physical requirements to perform this repetitive dynamic movement, the likelihood of injury may increase [3].

The predominant technical gesture in golf is the swing, which comprises four phases: the set-up, backswing, downswing and followthrough. The set-up is the correct alignment of the golfer with the ball; the backswing consists of the preparatory movements that take the club away from the ball, resulting from the rotation of the shoulder and hip around a fixed base of support; the downswing is the descending sequence from the top of the backswing towards the ball with maximum speed; and the followthrough consists of decelerating the club [4], using eccentric muscular actions where large rotational forces are transferred to the golf ball with compression loads at the bottom of the spine up to 7/8 times body weight [5].

Golf injuries can occur from overuse or trauma and mainly affect the elbow, wrist, shoulder and lumbar spine [5]. That said, injuries that occur in golf should not be underestimated, as they can have a major impact on a golfer’s career, including absenteeism from the sport, reduced performance and loss of profit [6].

Since there are still few studies on Portuguese championships, this study aimed to investigate the epidemiology of musculoskeletal injuries in golf athletes who participated in championships in the Algarve region, southern Portugal, determining the anatomical locations, types and mechanisms of most frequent injuries, as well as the associated risk factors.

## 2. Materials and Methods

This study was approved by KinesioLab–Research Unit in Human Movement, Piaget Institute (KL.2023.05), and this was a descriptive-correlational and cross-sectional study.

This study was conducted in accordance with the Declaration of Helsinki. To participate in the study, all athletes signed a consent form stating that they agreed to participate in the study, the nature and objective of which was explained to them, and it was also clarified that they could withdraw from the study at any time without having give some explanation and without any type of prejudice. The data collected were treated anonymously, not associated with individual responses and used exclusively for research purposes.

### 2.1. Participants

The study population consisted of 234 golf athletes of all sexes and of any nationality, who participated in 5 championships in the Algarve region, southern Portugal, organized by the Portuguese Golf Federation.

The following competitions in the Algarve region were included: Palmares Open II–Palmares Golf Lagos, Dom Pedro Pinhal Open D3–Vilamoura, Pinhal, Palmares Classic–Palmares Golf Lagos, Vilamoura Tournament–Vilamoura Golf Club and OPTILINK TOUR CHAMPIONSHIP–Vilamoura, Laguna.

The inclusion criteria cumulatively included the following: golf athletes who had practiced the sport for at least twelve months, who were aged 18 years or over, who freely agreed to participate in the study, who signed the informed consent form and who were present on the days of data collection during competitions. The exclusion criteria excluded golf athletes who did not speak Portuguese or English.

### 2.2. Measuring Instruments

The measuring instrument used was a questionnaire, in digital format, adapted from the IGF consensus on reporting and recording injury in golf, BJSM [7].

The questionnaire was subjected to a pre-test involving a sample of 10 golf athletes with the same characteristics as the target population. There were no changes to the measurement instrument after carrying out the pre-test, but it served to account for the average response time taken to complete the questionnaire, which was 3.27 min. The main purpose of the pre-test was to clarify possible interpretation doubts, collect possible observations for improvement and check the completion time.

The questionnaire is divided into 3 parts: sociodemographic characteristics of the population and modality, occurrence of injury and characterization of the injury.

The first part of the questionnaire included the following questions about gender, age, nationality, years of playing the sport, whether the athlete was part of the golf federation, regularity of training per week, whether they trained more than once a day, their weekly training hours, dominant side, warming up before training/competitions and cooling down after finishing training/competition.

In the second part of the questionnaire, the athlete was asked about injury presence throughout golf practice and, if the answer was negative, the questionnaire ended. If positive, the athlete was asked about the number of injuries they suffered during this period. Next, the athlete was asked if he/she had any injuries related to golf during the following periods: at the time of applying the questionnaire (at the current moment), in the last 6-month and in the 12-month periods. Only athletes who reported injuries in the last 12 months continued to fill out the questionnaire.

An injury was defined as any condition or symptom that occurred as a result of sport practice (training and competition) and had at least one of the following effects: the practitioner had to interrupt training/competition for at least one day; the practitioner did not have to stop the sport but had to modify it (for fewer hours of practice or training, lower intensity of effort, or less able to perform certain gestures or movements/techniques); the professional sought advice or treatment from healthcare professionals to treat the condition or symptom [8].

In the third part of the questionnaire, the athlete identified the number of injuries suffered in the 12-month period, being able to characterize a maximum of three injuries, regarding the type and location of the injury, the occurrence of the injury, whether any treatment was performed and, if so, they were asked to mention the type of treatment carried out, the mechanism of injury, the downtime resulting from each injury and the current situation of each injury.

Data collection took place in person, by physiotherapy students from the Escola Superior de Saúde Jean Piaget do Algarve, in golf championships that took place in the Algarve region.

### 2.3. Data Analysis

The software used to perform the statistical analysis of the data was the Statistical Package for Social Sciences (SPSS), version 28.0. 

In the first approach, descriptive statistics were performed. To determine the injury proportion, the total number of participants who had at least one injury in the last 12 months was divided by the total number of athletes. The injury rate value refers to the total number of injuries divided by the total time the golfer is exposed to risk (defined as 1000 h). This total time of injury risk was calculated by multiplying the average total hours of training by the frequency of training, both over a period of one week, and this value was multiplied by 12 months (52 weeks). The injuries average number per athlete was calculated by division of the total injuries number by the total sample number of athletes. The average of injuries per injured athlete was calculated by division of the total injuries number by the total number of injured athletes [9].

Binary logistic regressions (Enter methods) were applied to test the influence of the variables used in this study on the injury presence. The statistical significance level was established at 0.05. 

## 3. Results

The sample consisted of 140 athletes, corresponding to 60% of the athletes who participated in the competitions, aged between 18 and 72 years (29.6 ± 13.5 years), being the majority (133; 95%) male and 7 (5%) female athletes. The majority of athletes (96; 68.6%) belonged to a federation in his/her country, and most had the right side as their dominant side (127; 90.7%).

Argentina, Canada, Spain, Estonia, Finland, France, Indonesia, Switzerland and the United States of America were represented by 1 (0.7%) athlete each. Czech Republic, Denmark and Italy were represented by 2 (1.4%) athletes. South Korea had 3 (2.1%) athletes. Slovenia and Sweden by 4 (2.9%) athletes. Netherlands by 5 (3.6%) athletes, Germany by 8 (5.7%), Ireland by 11 (7.8%), Portugal by 16 (11.4%) and United Kingdom by 75 (53.6%) athletes. 

Most athletes had practiced golf for more than 10 years (104; 74.3%), 4 (2.8%) had practiced between 1 and 2 years, 5 (3.6%) between 3 and 4 years, 5 (3.6%) between 5 and 6 years, 7 (5%) between 7 and 8 years and 15 (10.7%) between 9 and 10 years. The weekly training frequency ranged from 1 to 7 times per week (5.1 ± 1.7), with most athletes training more than once a day (80; 57.1%). The weekly training duration varied between 1 and 60 h (26.8 ± 17.1). Almost all athletes (133; 95%) warmed up before training and competitions, with a duration that varied between 5 and 90 min (26.9 ± 24.1), but cooling down was performed by fewer athletes (63; 45%), with a duration between 4 and 60 min (13.6 ± 14.9).

Table 1 shows the presence of injuries in athletes in all periods questioned in the survey. During the entire golf practice, 32 (45.7%) athletes reported suffering 1 injury, 24 (34.3%) athletes suffered 2 injuries, 3 (4.3%) suffered 3 injuries and 11 (15.7%) suffered 4 or more injuries, totaling 133 injuries. Within the last 12-month period, 29 (67.4%) athletes reported suffering 1 injury, 8 (18.6%) athletes suffered 2 injuries, 4 (9.3%) athletes suffered 3 injuries and 2 (4.7%) suffered 4 or more injuries, totaling 65 injuries.

Table 2 shows the type and anatomical local of the injuries. The number of total injuries shown in this table is less than the total number of injuries in the 12-month period because only a maximum of 3 injuries were classified per athlete.

Table 3 presents the absolute and relative frequencies of characterization of the injuries over a 12-month period.

The value of injury proportion was 0.31 (CI 95%: 0.45–1.07), and the injury rate was 0.33 injuries per 1000 h of golf training. The average number of injuries per athlete was 0.46. The average of injuries per injured athlete was 1.51.

Regarding treatment, 56 (88.9%) injuries were treated and 7 (11.1%) were not. Most injuries were treated with physiotherapy (48; 38.7%), followed by rest (30; 24.2%), medication (18; 14.5%), osteopathy (13; 10.5%), immobilization (6; 4.8%), unconventional therapies (6; 4.8%) and surgery (3; 2.4%). The total number of types of treatments (*n* = 124) is greater than the number of injuries (*n* = 63) because some athletes performed more than one type of treatment.

Table 4 shows the relationship between the injury presence in a 12-month period and the non-modifiable sample factors (sex and age group) and the golf practice characteristics variables analyzed in this study. The only relationship that achieved statistical significance with injury presence was the frequency of weekly training, in which it was observed that athletes who train 4 times or more per week were 3.5 times more likely to develop an injury while playing golf.

## 4. Discussion

Although golf is considered a sport with a minimal risk of injuries, as it is not a sport that presents a lot of impact, contact with other athletes and high intensity levels of strength, power and speed, data from this study revealed a moderate prevalence of injuries in the athletes who made up the analyzed sample. Similar results were obtained in the Fradkin et al. [6] study that evaluated 522 female golfers that participated in competition in Australia and verified that 35.2% of the golfers reported having sustained a golfing injury within the previous 12 months, totaling 184 injuries. Another study of Fradkin et al. [10] evaluated 304 golfers and verified the presence of injuries in 36.5%, reporting 111 injuries over a 12-month period. Qureshi et al. [11] evaluated 76 amateur golfers, and injuries were reported by 45% of golfers during a six-month period. 

Gosheger et al. [12] obtained higher values, for a sample of 60 professional golfers, where 36 (60.0%) athletes suffered 110 injuries, an average of 3.06 injuries per player, and, for 643 amateur golfers, 255 (39.7%) reported 527 injuries, an injury rate of 2.07 injuries per player. The data were collected in Germany during two golfing seasons (1999 and 2000). Walsh et al.’s [13] study analyzed golf-related injuries treated in US hospital emergency departments from 1990 and verified that 663,471 individuals equal to or more than 7 years old were treated in this emergency department for golf-related injuries, averaging 30,158 annually or 12.3 individuals per 10,000 golf participants, in which patients 18–54 years old registered 42.2% of injuries.

The anatomical sites that suffered the most injuries in this study were the lumbar spine (27%) and the wrist (22%). Several studies also revealed the lumbar spine as the most affected site [6,10]. Qureshi et al. [11] revealed that 29% of injuries were to the wrist and 21% to the back. In the study of Gosheger et al. [12], 16.3% of injuries occurred in the lumbar spine and 14.1% in the wrist, with the most affected sites being the elbow (22.3%) and shoulder (17.6%). However, data from Walsh et al. [13] revealed that the head/neck was the most frequently injured body region (36.2%), and in Joeng et al.’s study [14], the most common locations were the shoulder/clavicle.

McHardy and Pollard [15] carried out a review of the epidemiological literature on low back pain in golfers and showed that the incidence of golf-related low back injury ranges from 15 to 34% in the amateur golfer and 22 to 24% in the professional ranks. The lumbar spine is an area of the body that undergoes significant movement and muscular activity during the swing movement, and this significant and repetitive activity may be associated with the high rate of injuries in golfers [15]. 

The downswing phase begins with a hip movement, and the following phase is characterized by a body position known as “reverse C” (modern swing). This position is characterized by an imaginary line drawn from the right foot to the player’s neck and head that is shaped like an inverted “C”. This position creates a large degree of hyperextension in the lumbar spine and has been proposed as a mechanism for many of the overuse injuries seen in the lower back of golfers [15,16,17]. The classic golf swing emphasizes reducing the hip–shoulder separation angle by elevating the front heel during the swing to increase hip turn, which results in reduced torque on the spine. With the modern swing, the golfer will experience increased lateral flexion and exaggerated hyperextension of the back during the followthrough [18]. Nowadays, many golfers utilize the modern swing because it allows for longer range and higher accuracy [13].

A golf swing involves a number of loading patterns, including lateral bending, compression (more than 8 times body weight), shear forces (translational) and axial rotation [11,15,17,18]. A golf swing also contributes to lower back injuries and muscle imbalances due to its asymmetrical nature [18,19]. The risk of developing low back pain can increase when there are technical failures and a lack of adequate strength and mobility. Inadequate loading resulting from poor technique can damage soft tissue, causing degeneration of skeletal joints over time. Low back pain can also occur from overuse due to the volume of practice and play and decreased variability in the swing, which increases the cyclical nature of musculoskeletal loading [16,19]. There is a hypothesis that modifying the golf swing (classic swing and hybrid swing) may reduce the incidence of low back pain in golf. More research needs to be performed on the various golf swings to evaluate whether different swings can change low back injury rates in golfers [15].

Regarding wrist movement during golf practice, the wrists move with a wide range of motion to execute a proper swing [18]. In the final movement of the backswing, the wrists are in radial deviation, with one of the wrists in sub maximal extension [15]. Wrist injuries can occur due to hitting the shot, hitting the ball “fat” (i.e., hitting the ground before the ball), when hitting firm ground or when hitting very heavy rough terrain, due to excessive use [15,18,20], after the impact of the ball during the initial follow-up, when the driving wrist suffers ulnar deviation and supination or by a sudden and strong impact to the wrist resulting from hitting the ground before the ball [15,18].

The most common types of injuries in our study were muscle sprain or rupture (31%) and low back pain (18%). Several studies [6,10] also found strain to be the most common injury, Walsh et al. [13] verified that sprain/strain (30.6%) was the most common type of injury and in Joeng et al.’s [14] study it was tendinosis or tendinopathy (21.2%).

The most frequent injury mechanisms reported by our study sample were repetitive movements and rotational movement (a single movement). Similar data were found in other studies, which presented repeated movement as the main injury mechanism [6,10,12]. However, Walsh et al.’s [13] study revealed that the most common mechanisms of injury were injured by a golf club (23.4%) or struck by a golf ball (16.0%). 

Professional golfers typically hit more than 2000 balls per week, with 73.3% striking 200 balls or more per day on average and only 19.4% of amateurs hitting more than 200 balls per week [21]. In both cases, that represents a high repetition load of the same type of movement. Strains can be the result of overuse (prolonged, repetitive movement) of muscles and tendons.

Regarding the time lost to recover from injuries, most injuries in our study had a brief recovery period (3 to 7 days), differing from the study by Gosheger et al. [12] in which the total loss was 18,221 days to 637 golf injuries, for an average of 28 days or 4 weeks per injury. In Fradkin et al.’s [10] study, the majority of injured golfers needed treatment for 1 to 2 weeks (29.8%), and 51.3% golfers reported an impact on their lives.

The type of treatment that the injured golfers in our sample resorted to was physiotherapy, showing data similar to those obtained in the study of Fradkin [6], and that the most common health professional visited for treatment was a physiotherapist (20.1%).

Regarding risk factors, the frequency of weekly training was the unique variable that presented a statistical relationship with injury presence, which is in line with the other results obtained, since a greater frequency of training is associated with a greater repetition of movements and the type of injury found in this study.

Our study showed some limitations that are common in other retrospective cross-section studies that rely on the accurate recall of events by the individuals and a paucity of standardized medical documentation. Future studies should indicate the relationship between injuries observed and the golf season, comparing the frequency of injuries in tournaments that occurred at the beginning, course and end of the season, in order to relate injuries to the overload of training or competitions. Future studies could also relate athletes’ handicap to the presence of injuries. Epidemiological studies on golfers are still scarce when compared to other sports, making it necessary to carry out future studies involving amateur and competitive golfers of various ages and nationalities.

Knowledge and understanding of the golf practice characteristics help to outline prevention strategies, such as the emphasis on movement correction, the importance of pauses and intervals between training, respecting a recovery period and the implementation of training to improve muscular strength and flexibility in order to promote a greater stability to non-contractile structures, among other aspects.

## 5. Conclusions

Our study evaluated the epidemiology on golf injuries and showed a moderate presence of injury in the analyzed sample, in which the most common anatomical body locations were lower back and wrist, the majority of injuries were sprain or rupture and low back pain and the most frequent injury mechanisms reported were repetitive movements and rotational movement (a single movement), with training frequency equal to or more than 4 times per week being the risk factor associated with injury presence in golf practice.

## Figures and Tables

**Table 1 ijerph-21-00542-t001:** Presence of injury in golf athletes.

Injury Presence							
Throughout practice		At the moment		6-month period		12-month period	
Absence	Presence	Absence	Presence	Absence	Presence	Absence	Presence
70 (50.0%)	70 (50.0%)	120 (85.7%)	20 (14.3%)	104 (74.3%)	36 (25.7%)	97 (69.3%)	43 (30.7%)

**Table 2 ijerph-21-00542-t002:** Type and location of injury in a 12-month period.

Type of Injury	Location of Injury	*n*	%
Muscle sprain or rupture	Cervical spine	2	
	Lumbar spine	5	
	Thorax/chest/ribs	3	
	Shoulder	1	
	Wrist	3	
	Pelvis	3	
	Knee	2	
	All	19	30.9%
Low back pain	Lumbar spine	11	
	All	11	17.5%
Non-specific pain	Cervical spine	1	
	Thorax/chest/ribs	1	
	Wrist	3	
	Knee	2	
	Foot and fingers	1	
	All	8	12.7%
Ligament injury	Elbow	1	
	Wrist	2	
	Knee	1	
	Ankle	2	
	All	6	9.5%
Tendinopathy	Shoulder	3	
	Elbow	1	
	Wrist	2	
	All	6	9.5%
Muscle contusion	Cervical spine	1	
	Lumbar spine	1	
	Shoulder	1	
	Wrist	2	
	Thigh	1	
	All	6	9.5%
Fracture	Wrist	1	3.2%
	Thorax/chest/ribs	1	
	All	2	
TOTAL	63		100%

**Table 3 ijerph-21-00542-t003:** Injuries characterization in a 12-month period.

Characterization of the Injury		*n*	%
Anatomical location	Cervical spine	4	6.3
	Lumbar spine	17	27.0
	Thorax/chest/ribs	5	7.9
	Shoulder	5	7.9
	Elbow	3	4.8
	Wrist	14	22.2
	Pelvis	3	4.8
	Thigh	1	1.6
	Knee	5	7.9
	Foot and fingers	1	7.9
Occurrence of the injury	During training	30	47.6
	During competition	24	38.1
	During cool-down	5	7.9
	During warm-up	4	6.3
Mechanism of injury	Repetitive movements	42	66.7
	Rotational movement (a single movement)	7	11.1
	Direct contact with some object (some part of the athlete’s body touches or is hit by some object)	6	9.6
	Prolonged effort	5	7.9
	Fall	3	4.8
Downtime resulting from injury	No day, conditionally	13	20.6
	Up to 2 days	6	9.5
	Between 15 and 30 days	5	7.9
	Between 3 and 7 days	18	28.6
	Between 8 and 14 days	8	12.7
	Between 15 and 30 days	4	6.3
	More than 30 days	9	14.3
Current situation of injury	No pain or other symptoms and fully recovered	35	55.6
	No pain or other symptoms but still being treated or conditioned in practice	17	27.0
	With pain or other symptoms and being treated	8	12.7
	With pain or other symptom and without treatment	3	4.7

**Table 4 ijerph-21-00542-t004:** Relationship between the presence of injury and variables analyzed in this study.

Variables	OddsRatio_crude_ (CI 95%)	*p*-Value
Sex (female *) male	2.77 (0.32–23.73)	0.35
Age group (≥34 years old *) 18 to 33 years old	1.16 (0.55–2.42)	0.69
Years of practice (>10 years *) until 10 years *)	1.65 (0.74–3.65)	0.22
Weekly training (until 3 times *) ≥ 4 times	3.46 (0.97–12.36)	0.05
Duration of training per week (until 30 h *) > 30 h	1.79 (0.87–3.72)	0.12

* Class reference.

## Data Availability

The data obtained in this study are included in an SPSS database. None of these documents are available online.

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
