# Peer review of "Epidemiology of Musculoskeletal Injuries in Golf Athletes: A Championship in Portugal"

_ijerph, 2024, doi:10.3390/ijerph21050542_

Round 1

Reviewer 1 Report

Comments and Suggestions for Authors

This manuscript studies the epidemiology of golfers' injuries in tournaments held in Portugal. The approach is based on surveys of a large sample of participants and the impact of injuries in this context is assessed.

In my view, the approach of the study is correct, although some remarks can be made to improve the manuscript:

In general, major problems only concern the categorization of the players, while I find no serious problems regarding the methodology of the study, the presentation of the results or the written tables. In this work it is not necessary to have control subjects because it is a descriptive study.

Among the major problems, they should indicate relevant aspects about the technical level of the participants in the study, such as the golf handicap. In relation to the handicap, they should indicate the correlation between the injuries observed, both in terms of type and location, with respect to this index of level of play. Likewise, they should correlate handicap with gender, age and hours of practice.

They should also indicate the relationship of the injuries observed with the playing schedule of the golf season. They should indicate whether the tournaments evaluated correspond to the beginning, course or end of the season in order to relate the injuries to the overload of training or competitions.

Regarding the injuries, they should indicate the treatments that the participants followed and that could have an impact on the time in which they suffered the injury.

They should also indicate the relationship of the injuries observed with the playing schedule of the golf season. They should indicate whether the tournaments evaluated correspond to the beginning, course or end of the season in order to relate the injuries to the overload of training or competitions.

With regard to injuries, it would be necessary for them to indicate whether they occurred during the golf tournament under study or whether they were already injured at the time they started their participation. In both cases they should indicate the impact on the study and their limiting factors.

Author Response

Dear reviewer,

We appreciate the time, and effort you have dedicated to providing insightful feedback on ways to strengthen our paper. To facilitate your review, the following is a point-by-point response to the questions and comments.

This manuscript studies the epidemiology of golfers' injuries in tournaments held in Portugal. The approach is based on surveys of a large sample of participants and the impact of injuries in this context is assessed.

In my view, the approach of the study is correct, although some remarks can be made to improve the manuscript:

In general, major problems only concern the categorization of the players, while I find no serious problems regarding the methodology of the study, the presentation of the results or the written tables. In this work it is not necessary to have control subjects because it is a descriptive study.

Among the major problems, they should indicate relevant aspects about the technical level of the participants in the study, such as the golf handicap. In relation to the handicap, they should indicate the correlation between the injuries observed, both in terms of type and location, with respect to this index of level of play. Likewise, they should correlate handicap with gender, age and hours of practice.

Answer: The handicap was not assessed in this study, only the years of sports practice, this variable being related to the presence of injury. The handicap is the assessment of a player's level (more precisely, his potential at a given moment) and which will allow that player an additional number of shots to reach Par on a course. As the level of play improves, the handicap naturally drops. The determination of the handicap is done through a mathematical model that works on the results obtained by the player on the different fields in which he plays.

They should also indicate the relationship of the injuries observed with the playing schedule of the golf season. They should indicate whether the tournaments evaluated correspond to the beginning, course or end of the season in order to relate the injuries to the overload of training or competitions.

Answer: It would be a great objective for a future study, we will add this suggestion in the discussion section in the suggestions for future studies. In this study it will not be possible to answer this question, as not all participants competed in all tournaments, so data was collected in several tournaments to obtain a larger sample of players. A player who had competed in one of the tournaments would not answer the questionnaire again if they participated in another tournament.

We adder this paraghrafh in the end of discussion section: “Our study showed some limitations that are common in other retrospective cross-section studies that rely on the accurate recall of events by the individuals and a paucity of standardised medical documentation. Future studies should indicate the relationship between injuries observed and the golf season, comparing the frequency of injuries in tournaments that occurred at the beginning, course and end of the season, in order to relate injuries to the overload of training or competitions.

Regarding the injuries, they should indicate the treatments that the participants followed and that could have an impact on the time in which they suffered the injury.

Answer: The types of treatment were mentioned, but no questions were asked about the impact of this treatment on improving the injury, as it is beyond the objective of the study. What was questioned is the current situation of the injury and the downtime caused by the injury, which is indirectly related to the type of treatment performed. However, it is not possible to make any association between these variables, as in addition to not meeting the objective of the study, individuals could undergo more than one treatment for just one injury, making this association with the evolution of the injury impossible, which I mention again was not questioned.

They should also indicate the relationship of the injuries observed with the playing schedule of the golf season. They should indicate whether the tournaments evaluated correspond to the beginning, course or end of the season in order to relate the injuries to the overload of training or competitions.

Answer: It has already been answered.

With regard to injuries, it would be necessary for them to indicate whether they occurred during the golf tournament under study or whether they were already injured at the time they started their participation. In both cases they should indicate the impact on the study and their limiting factors.

Answer: We did not question whether the injuries occurred during the tournament in which the data were collected, this would be another objective of another study (epidemiological data on injuries that occurred during golf tournaments). To do this, it would be necessary to have a medical record of injuries occurring in golf championships, which we do not have access to and, as we said, would be another type of study. Regarding the fact that the athletes were already injured, we asked whether they were already injured at the time they started participating (see table 1).

“In both cases, they must indicate the impact on the study and its limiting factos”. - I didn't understand your suggestion, but as I already explained, these suggested variables were not carried out in this study.

Reviewer 2 Report

Comments and Suggestions for Authors

Dear authors,

Thank you for the opportunity to review you paper. Overall, the paper requires a detailed proof-read and improvements in English writing.

I have included comments on each section below: 

ABSTRACT

- Needs significant grammatical / English language corrections. 

- There is no conclusion - what are the implications of your findings. Why are they useful? 

- It would be helpful to include the level of championship and players in the abstract. 

INTRODUCTION

Clear - but again has no rationale as to why we need to know this information.

I would like to see more detail in the mechanisms of injury. 

You state there is no data specifically from Portugal - but is there from other countries? If so, add these to the introduction. Why do we need to have data specifically from Portuguese championships? 

METHOD

Please provide the ethics approval reference number. 

Line 56, page 2 is overly long and unclear. 

What level of competition are these? Do you need to be a certain "level" to compete? 

What were the outcomes from the pre-test - any changes made? 

Line 86 page 2 - they are not questions. 

Line 92 and 94 page 2 - "his" and "he" but both genders are included?

Why were injuries less than 12-months old excluded? 

Line 114 - "static analysis" - statistical?

Line 119 - surfer?

More detail on the logistic regression method is required. 

RESULTS

The UK and England are the same place - not sure why these are reported separately. The characteristics of participants, including nationalities, should be included in a clear table, rather than confusing text. 

The timing of injury occurrence is unclear -is Throughout practice, this practice? Or 12 months of practice?  

What is the time frame for Table 2 and how is it different to Table 3? 

As the number of females was so low, it is not appropriate to use it in your regression model. 

The p-value for weekly training is ABOVE 0.05, and hence, not significant. Your following discussion needs to reflect this. 

I would like to see a re-work of the results section as at the moment it is very confusing. 

DISCUSSION

Line 184 - how are you defining moderate? 

Comparisons to other studies have little context as you have not provided the level of your players (amateur / professional). 

More application and implication of your findings is needed - you mention some in the conclusion but they are not discussed earlier. 

Comments on the Quality of English Language

This work needs a thorough proof-read for grammatical and spelling errors. At the moment, the language is confusing. 

Author Response

Dear reviewer, 

We appreciate the time, and effort you have dedicated to providing insightful feedback on ways to strengthen our paper. To facilitate your review, the following is a point-by-point response to the questions and comments. 

Dear authors, 

Thank you for the opportunity to review you paper. Overall, the paper requires a detailed proof-read and improvements in English writing. 

I have included comments on each section below:  

ABSTRACT 

- Needs significant grammatical / English language corrections.  

- There is no conclusion - what are the implications of your findings. Why are they useful?  

Answer: This is a short conclusion “It was observed moderate injury presence, being the high training frequency, a risk factor associated.” 

We already have 200 words in the abstract, I can't yet write the implications, which I think are not more important than the rest of the information that is written. 

- It would be helpful to include the level of championship and players in the abstract.  

Answer: The handicap was not assessed in this study, only the years of sports practice, this variable being related to the presence of injury. The handicap is the assessment of a player's level (more precisely, his potential at a given moment) and which will allow that player an additional number of shots to reach Par on a course. As the level of play improves, the handicap naturally drops. The determination of the handicap is done through a mathematical model that works on the results obtained by the player on the different fields in which he plays. 

INTRODUCTION 

Clear - but again has no rationale as to why we need to know this information. 

Answer: The 1st paragraph presents a justification for carrying out the study, specifically the choice of the sport of golf to be studied, as it has a great impact on the economy.The 2nd paragraph relates the practice of golf with the possible occurrence of injuries. The 3rd paragraph briefly describes the practice of golf, describing concepts for the reader who has no knowledge in the area and also relates golf gestures to the possible occurrence of injuries. The 4th paragraph again presents a justification for the importance of carrying out this type of study, in order to provide data to professionals to create injury prevention strategies. The last paragraph again presents a justification for the development of this study (in the absence of national studies) and presents the objective of the study. Could the reviewer be more specific about what could be improved? An introduction basically has to present definitions of concepts, justifications for carrying out the study and could present study results and, if there were divergences between study results, it could present them in order to have another way of justifying the study. As there are already few studies on the epidemiology of golf injuries, we decided not to present these data in the Introduction but rather in the discussion of the article. 

I would like to see more detail in the mechanisms of injury.  

Answer: The mechanisms of injury are explained in detail in the discussion of the article. If this information were added to the introduction, it would be repeated, and for us it makes more sense to better describe the mechanisms in the discussion of the article. Furthermore, some injury mechanisms are mentioned in the introduction. 

“movement of muscles throughout a high range of motion” “repetitive dynamic movement” “large rotational forces are transferred to the golf ball with compression loads at the bottom of the spine up to 7/8 times body weight” 

You state there is no data specifically from Portugal - but is there from other countries? If so, add these to the introduction. Why do we need to have data specifically from Portuguese championships?  

Answer: There are data from other countries, but they are also few and, as well as the injury mechanisms, we chose to add the results of other studies in the discussion of the article in order to compare them with our results and to avoid having repeated information in the introduction and in the discussion. About having data from Portuguese championships, because the study was carried out in Portugal, and we want to have data from championships in this country. 

METHOD 

Please provide the ethics approval reference number.  

Answer: Done. 

Line 56, page 2 is overly long and unclear.  

Answer: Done. 

What level of competition are these? Do you need to be a certain "level" to compete?  

Answer: As said before, the handicap was not assessed in this study and we do not describe level. 

What were the outcomes from the pre-test - any changes made?  

Answer: We added this: The questionnaire was subjected to a pre-test involving a sample of 10 golf athletes with the same characteristics as the target population. There were no changes to the measurement instrument after carrying out the pre-test, but it served to account for the average response time taken to complete the questionnaire, which was 3.27 minutes. 

Line 86 page 2 - they are not questions.  

Answer: We added the word about: The first part of the questionnaire included the following questions about: 

Line 92 and 94 page 2 - "his" and "he" but both genders are included? 

Answer: We change for this: In the second part of the questionnaire, the athlete was asked about the injury presence throughout the athlete golf practice and, if the answer was negative, the questionnaire ended. If so, the athlete was asked about the number of injuries they suffered during this period. Next, the athlete was asked whether he/she had suffered 

Why were injuries less than 12-months old excluded?  

Answer: Because we evaluate the period of injury in the last 12 months, therefore athletes could not have less than 12 months of practice as the 12-month period would not be evaluated. It was the injuries from the last 12 months that were characterized in terms of type, location, etc. 

Line 114 - "static analysis" - statistical? 

Answer: Adjusted. 

Line 119 - surfer? 

Answer: Adjusted. 

More detail on the logistic regression method is required.  

RESULTS 

The UK and England are the same place - not sure why these are reported separately.  

Answer: And Scotland too. Both were added in the UK. Thanks. 

The characteristics of participants, including nationalities, should be included in a clear table, rather than confusing text.  

Answer: The article already has many tables and we like to present results in tables that directly respond to the objective of the study. Nationality is just a characterization of the sample, so we think it is not appropriate to present this data in a table. 

The timing of injury occurrence is unclear -is Throughout practice, this practice? Or 12 months of practice?   

Answer: It was described in the methodology, but we changed the phrase to see if it becomes more noticeable. “. Next, the athlete was asked if he/she had any injuries related to golf during the following periods: at the time of applying the questionnaire (at the current moment), in the last 6-months and in the 12-months periods.” 

What is the time frame for Table 2 and how is it different to Table 3?  

Answer: We added the period in the legend (“in a 12-month period”), but all the characterization of the injuries arise from the 12-month period.  

As the number of females was so low, it is not appropriate to use it in your regression model.  

Answer: We added women in the logistic regression, since this statistical test does not consider proportions in its analysis (like Chi-square test, for exemple), so the reduced number of women would not affect the analysis. Furthermore, we wanted to understand which sex would be more likely to suffer injuries. If we excluded women from the regression, we should also exclude from all results obtained and include in the methodology that women were excluded, without having a justification for this. 

The p-value for weekly training is ABOVE 0.05, and hence, not significant. Your following discussion needs to reflect this.  

Answer: This is because the 4th decimal place was rounded, we have already corrected the value. Thanks 

I would like to see a re-work of the results section as at the moment it is very confusing.  

Answer: The results section begins by presenting data on the characterization of the sample, descriptive data on all variables, the data that best respond to the objectives are presented in tables and finally we present the results of inferential statistics. The tables do not completely repeat information that is written in the text. Could you explain to me what is confusing, as this is the correct order in a scientific methodology of how to present the results of an article. 

DISCUSSION 

Line 184 - how are you defining moderate?  

Answer: You're right, there is no specific definition for moderate. Low could be considered up to 20% and high above 50%, therefore we define moderate as between 30% and 40%. Just writing that there was a presence of injury to the athletes also doesn't seem appropriate, which is why we opted for the word moderate. Any suggestion? 

Comparisons to other studies have little context as you have not provided the level of your players (amateur / professional).  

Answer: As already mentioned, we do not question the athletes' handicap. We add this as a suggestion for future studies. Thanks.  

“Future studies could also relate athletes' habdicap to the presence of injuries.” 

More application and implication of your findings is needed - you mention some in the conclusion but they are not discussed earlier.  

Answer: We believe that the implications of the study are best described in the conclusion of the article and should not be reinforced or repeated in the discussion. 

Comments on the Quality of English Language 

This work needs a thorough proof-read for grammatical and spelling errors. At the moment, the language is confusing.  

Answer: A review of the entire text was carried out.